# Hallo2: Long-Duration and High-Resolution Audio-Driven Portrait Image Animation

**Jiahao Cui**[1*], **Hui Li**[1*], **Yao Yao**[3], **Hao Zhu**[3], **Hanlin Shang**[1], **Kaihui Cheng**[1], **Hang Zhou**[2]
**Siyu Zhu**[1†], **Jingdong Wang**[2]
[1]Fudan University   [2]Baidu Inc.   [3]Nanjing University
jiahaocui279@gmail.com, 24110240042@m.fudan.edu.cn,
{yaoyao, zh}@nju.edu.cn, 24210240284@m.fudan.edu.cn,
chengkaihui1999@126.com, {zhouhang09, wangjingdong}@baidu.com
siyuzhu@fudan.edu.cn

## Abstract

Recent advances in latent diffusion-based generative models for portrait image animation, such as Hallo, have achieved impressive results in short-duration video synthesis. In this paper, we present updates to Hallo, introducing several design enhancements to extend its capabilities. First, we extend the method to produce long-duration videos. To address substantial challenges such as appearance drift and temporal artifacts, we investigate augmentation strategies within the image space of conditional motion frames. Specifically, we introduce a patch-drop technique augmented with Gaussian noise to enhance visual consistency and temporal coherence over long duration. Second, we achieve 4K resolution portrait video generation. To accomplish this, we implement vector quantization of latent codes and apply temporal alignment techniques to maintain coherence across the temporal dimension. By integrating a high-quality decoder, we realize visual synthesis at 4K resolution. Third, we incorporate adjustable semantic textual labels for portrait expressions as conditional inputs. This extends beyond traditional audio cues to improve controllability and increase the diversity of the generated content. To the best of our knowledge, Hallo2, proposed in this paper, is the first method to achieve 4K resolution and generate hour-long, audio-driven portrait image animations enhanced with textual prompts. We have conducted extensive experiments to evaluate our method on publicly available datasets, including HDTF, CelebV, and our introduced "Wild" dataset. The experimental results demonstrate that our approach achieves state-of-the-art performance in long-duration portrait video animation, successfully generating rich and controllable content at 4K resolution for duration extending up to tens of minutes.

## 1 Introduction

Portrait image animation—the process of creating animated videos from a reference portrait using various input signals such as audio Prajwal et al. (2020); Tian et al. (2024); Xu et al. (2024a); Zhang et al. (2023), facial landmarks Wei et al. (2024); Chen et al. (2024), or textual descriptions Xu et al. (2024b)—is a rapidly evolving field with significant potential across multiple domains. These domains include high-quality film and animation production, the development of virtual assistants, personalized customer service solutions, interactive educational content creation, and realistic character animation in the gaming industry. Consequently, the capability to generate long-duration, high-resolution, audio-driven portrait animations, particularly those assisted by textual prompts, is crucial for these applications. Recent technological advancements, notably in latent diffusion models, have significantly advanced this field.

Several methods utilizing latent diffusion models for portrait image animation have emerged in recent years. For instance, VASA-1 Xu et al. (2024b) employs the DiT model Peebles & Xie

---

[*]These authors contributed equally to this work.
[†]Corresponding Author

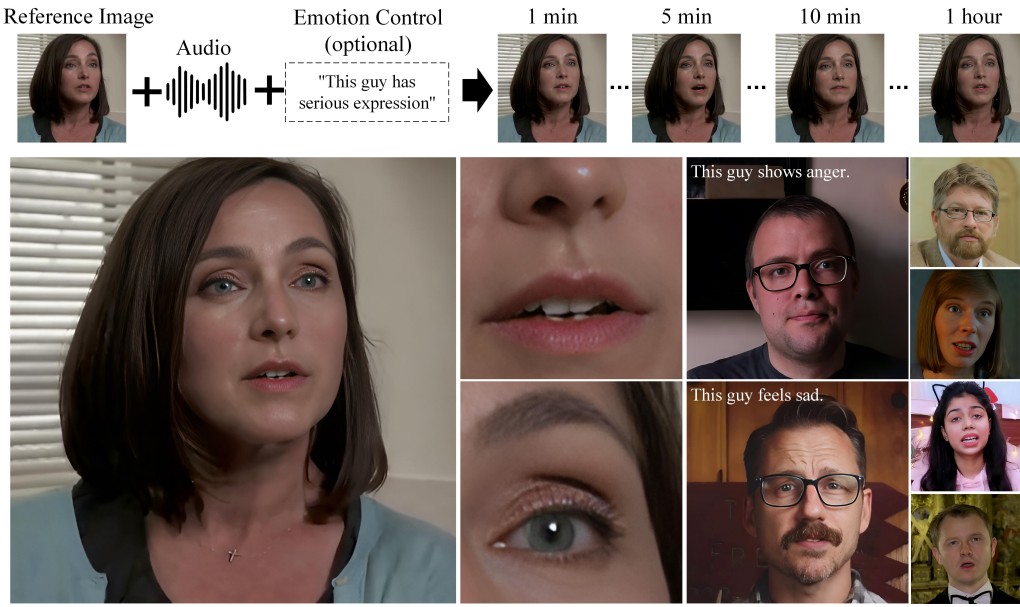

Figure 1: Demonstration of the proposed approach. This approach processes a single reference image alongside an audio input lasting several minutes. Additionally, optional textual prompts may be introduced at various intervals to modulate and refine the expressions of the portrait. The resulting output is a high-resolution 4K video that synchronizes with the audio and is influenced by the optional expression prompts, ensuring continuity throughout the extended duration of the video.

(2023) as a denoiser in the diffusion process, converting a single static image and an audio segment into realistic conversational facial animations. Similarly, the EMO framework Tian et al. (2024) represents the first end-to-end system capable of generating animations with high expressiveness and realism, seamless frame transitions, and identity preservation using a U-Net-based diffusion model Blattmann et al. (2023) with only a single reference image and audio input. Other significant advancements in this domain include AniPortrait Wei et al. (2024), EchoMimic Chen et al. (2024), V-Express Wang et al. (2024a), Loopy Jiang et al. (2024), and CyberHost Lin et al. (2024), each contributing to enhanced capabilities and applications of portrait image animation. Hallo Xu et al. (2024a), another notable contribution, introduces hierarchical audio-driven visual synthesis, building upon previous research to achieve facial expression generation, head pose control, and personalized animation customization. In this paper, we present updates to Hallo Xu et al. (2024a) by introducing several design enhancements to extend its capabilities.

Firstly, we extend Hallo from generating brief, second-long portrait animations to supporting duration of up to tens of minutes. As illustrated in Figure 2, two primary approaches are commonly employed for long-term video generation. The first approach involves generating audio-driven video clips in parallel, guided by control signals, and then applying appearance and motion constraints between adjacent frames of these clips Wei et al. (2024); Chen et al. (2024). A significant limitation of this method is the necessity to maintain minimal differences in appearance and motion across generated clips, which hampers substantial variations in lip movements, facial expressions, and poses, often resulting in blurriness and distorted expressions and postures due to the enforced continuity constraints. The second approach incrementally generates new video content by leveraging preceding frames as conditional information Xu et al. (2024a); Tian et al. (2024); Wang et al. (2021). While this allows for continuous motion, it is prone to error accumulation. Distortions, deformations relative to the reference image, noise artifacts, or motion inconsistencies in preceding frames can propagate to subsequent frames, degrading the overall video quality.

To achieves high expressiveness, realism, and rich motion dynamics, we follow the second approach. Our method primarily derives the appearance from the reference image, utilizing preceding generated frames solely to convey motion dynamics—including lip movements, facial expressions, and poses. To prevent contamination of appearance information from preceding frames, we implement a patch-drop data augmentation technique that introduces controlled corruption to the appearance information in the conditional frames while preserving motion characteristics. This approach en-

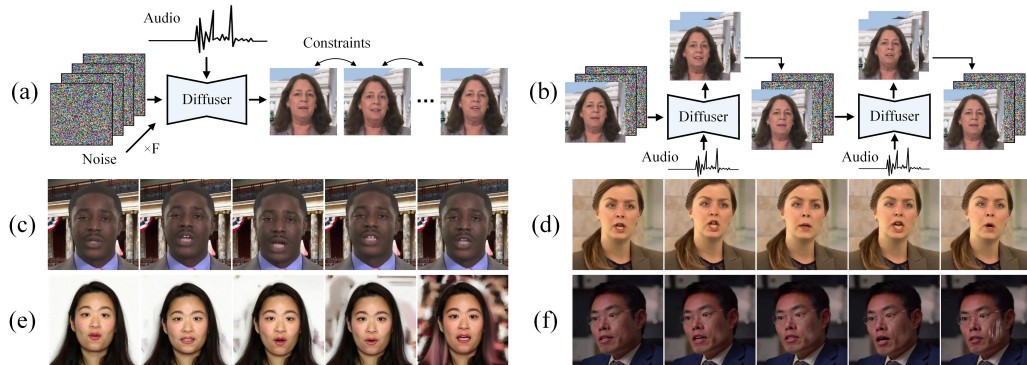

Figure 2: Comparison of parallel and incremental diffusion-based generative models for long-term portrait image animation. (a) Parallel generation. (b) Incremental generation. The parallel generation approach may lead to blurriness (c) and distorted expressions (d), due to inter-frame continuity constraints. The incremental generation method is susceptible to error accumulation in both backgrounds (e) and facial regions (f).

courages that the appearance is predominantly sourced from the reference portrait image, maintaining robust identity consistency throughout the animation and enabling long videos with continuous motion. Additionally, to enhance resilience against appearance contamination, we incorporate Gaussian noise as an additional data augmentation technique applied to the conditional frames, further reinforcing fidelity to the reference image while effectively utilizing motion information.

Secondly, to achieve 4K video resolution, we extend the Vector Quantized Generative Adversarial Network (VQGAN) Esser et al. (2021) discrete codebook space method for code sequence prediction tasks into the temporal dimension. By incorporating temporal alignment into the code sequence prediction network, we achieve smooth transitions in the predicted code sequences of the generated video. Upon applying the high-quality decoder, the strong consistency in both appearance and motion allows our method to enhance the temporal coherence of high-resolution details.

Thirdly, to enhance the semantic control of long-term portrait video generation, we introduce adjustable semantic textual prompt for portrait expressions as conditional inputs alongside audio signals. By injecting textual prompts at various time intervals, our method can help to adjust facial expressions and head poses, thereby rendering the animations more lifelike and expressive.

To evaluate the effectiveness of our proposed method, we conducted comprehensive experiments on publicly available datasets, including HDTF, CelebV, and our introduced "Wild" dataset. To the best of our knowledge, our approach is the first to achieve 4K resolution in portrait image animation for duration extending up to ten minutes or even several hours. Furthermore, by incorporating adjustable textual prompts that enable precise control over facial features during the generation process, our method ensures high levels of realism and diversity in the generated animations.

## 2 RELATED WORK

**Video Diffusion Models.** Diffusion-based models have demonstrated remarkable capabilities in generating high-quality and realistic videos from textual and image inputs. Stable Video Diffusion Blattmann et al. (2023) emphasizes latent video diffusion approaches, utilizing pretraining, fine-tuning, and curated datasets to enhance video quality. Make-A-Video Singer et al. (2022) leverages text-to-image synthesis techniques to optimize text-to-video generation without requiring paired data. MagicVideo Zhou et al. (2022a) introduces an efficient framework with a novel 3D U-Net design, reducing computational costs. AnimateDiff Guo et al. (2023) enables animation of personalized text-to-image models via a plug-and-play motion module. Further contributions, such as VideoComposer Wang et al. (2024b) and VideoCrafter Chen et al. (2023a), emphasize controllability and quality in video generation. CogVideoX Yang et al. (2024) enhances text-video alignment through expert transformers, and MagicTime Yuan et al. (2024) addresses the encoding of physical knowledge with a metamorphic time-lapse model. Building upon these advancements, our approach adopts superior pretrained diffusion models tailored specifically for portrait image animation, focusing on long-duration and high-resolution synthesis.

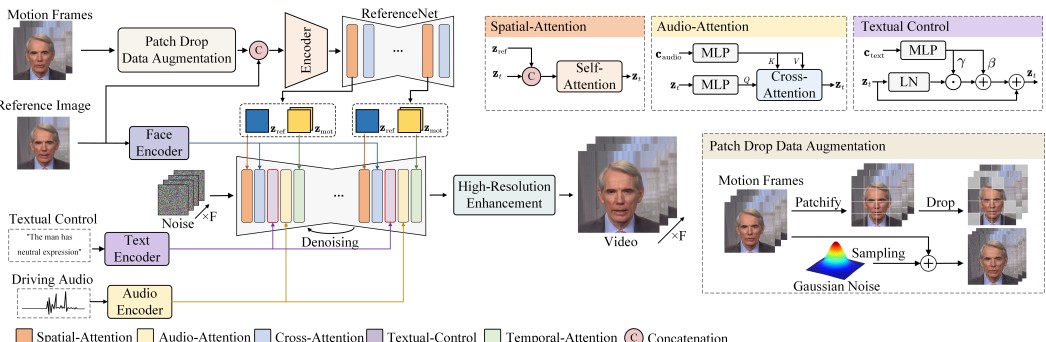

Figure 3: The framework of the proposed approach. The details of the proposed patch drop data augmentation is shown on the right side. $\mathbf{z}_{\texttt{ref}}$ and $\mathbf{z}_{\texttt{mot}}$ denote the features of the reference image and motion frames, respectively. $\mathbf{z}_t$ represents the latent features. $\mathbf{c}_{\texttt{audio}}$ and $\mathbf{c}_{\texttt{text}}$ correspond to the features of the audio condition and textual prompts, respectively.

**Portrait Image Animation.** Significant progress has been made in audio-driven talking head generation and portrait image animation, emphasizing realism and synchronization with audio inputs. Animate Anyone Hu et al. (2023) and MagicAnimate Xu et al. (2024c) utilize reference network and temporal attention modules to produce consistent and coherent human image animations. Lip-SyncExpert Prajwal et al. (2020) improved lip-sync accuracy using discriminators and novel evaluation benchmarks. Subsequent methods like SadTalker Zhang et al. (2023) and VividTalk Sun et al. (2023) incorporated 3D motion modeling and head pose generation to enhance expressiveness and temporal synchronization. ReSyncer Guan et al. (2024) revisited and rewired the Style-based generator to efficiently adopt 3D facial dynamics predicted by a principled style-injected Transformer. DiffTalk Shen et al. (2023) and DreamTalk Ma et al. (2023) improved video quality while maintaining synchronization across diverse identities. VASA-1 Xu et al. (2024b) and AniTalker Liu et al. (2024) integrated nuanced facial expressions and universal motion representations, resulting in lifelike and synchronous animations. AniPortrait Wei et al. (2024), EchoMimic Chen et al. (2024), V-Express Wang et al. (2024a), Loopy Jiang et al. (2024), CyberHost Lin et al. (2024), and EMO Tian et al. (2024) have contributed to enhanced capabilities, focusing on expressiveness, realism, and identity preservation. Despite these advancements, generating long-duration, high-resolution portrait videos with consistent visual quality and temporal coherence remains a challenge. Our method builds upon Hallo Xu et al. (2024a) to address this gap by achieving realistic, high-resolution motion dynamics in long-term portrait image animations.

**Long-Term and High-Resolution Video Generation.** Recent advances in video diffusion models have significantly enhanced the generation of long-duration, high-resolution videos. Frameworks like Flexible Diffusion Modeling Harvey et al. (2022) and Gen-L-Video Harvey et al. (2022) improve temporal coherence and enable text-driven video generation without additional training. Methods such as SEINE Chen et al. (2023b) and StoryDiffusion Zhou et al. (2024) introduce generative transitions and semantic motion predictors for smooth scene changes and visual storytelling. Approaches like StreamingT2V Henschel et al. (2024) and MovieDreamer Zhao et al. (2024) use autoregressive strategies and diffusion rendering for extended narrative videos with seamless transitions. In this paper, we employ patch-drop and Gaussian noise augmentation to enable long-duration portrait image animation.

Discrete prior representations with learned dictionaries have proven effective for image restoration. VQ-VAE Razavi et al. (2019) enhances VAEs by introducing discrete latent spaces via vector quantization, addressing posterior collapse. Building on this, VQ-GAN Lee et al. (2022) combines CNNs and Transformers to create a context-rich vocabulary of image components, achieving state-of-the-art results. CodeFormer Zhou et al. (2022b) uses a learned discrete codebook and employs a Transformer-based network for enhanced robustness against degradation. This paper proposes vector quantization of latent codes with temporal alignment techniques to maintain high-resolution coherence temporally for 4K synthesis.

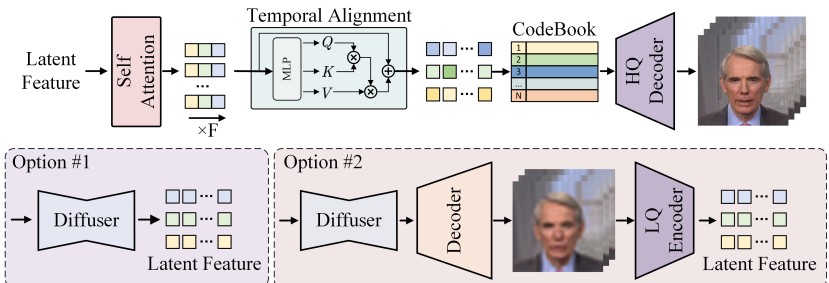

Figure 4: The illustration of the proposed high-resolution enhancement module. Two alternative designs for extracting input latent features are demonstrated.

## 3 METHOD

In this section, we introduce an extended technique for portrait image animation that effectively addresses the challenges of generating long-duration, high-resolution videos with intricate motion dynamics, as well as enabling audio-driven and textually prompted control. Our proposed method derives the subject's appearance primarily from a single reference image while utilizing preceding generated frames as conditional inputs to capture motion information. To preserve appearance details of the reference image and prevent contamination from preceding frames, we introduce a patch drop data augmentation technique combined with Gaussian noise injection (see Section 3.1). Additionally, we extend the VQGAN discrete codebook prediction into the temporal domain, facilitating high-resolution video generation and enhancing temporal coherence (see Section 3.2). Furthermore, we integrate textual conditions alongside audio signals to enable diverse control over facial expressions and motions during long-term video generation (see Section 3.3). Finally, we detail the network structure along with the training and inference strategies in Section 3.4.

### 3.1 LONG-DURATION ANIMATION

**Patch-Drop Augmentation.** To generate long-duration portrait videos that maintain consistent appearance while exhibiting rich motion dynamics, we introduce a patch drop data augmentation technique applied to the conditioning frames. The core idea is to corrupt the appearance information in preceding frames while preserving their motion cues, thereby ensuring that the model relies primarily on the reference image for appearance features and utilizes preceding frames to capture temporal dynamics.

Let $\mathbf{I}_{\text{ref}}$ denotes the reference image, and let $\{\mathbf{I}_{t-1}, \mathbf{I}_{t-2}, \ldots, \mathbf{I}_{t-N}\}$ represent the preceding $N$ generated frames at time steps $t-1$ to $t-N$. To mitigate the influence of appearance information from preceding frames, we apply a patch drop augmentation to each frame $\mathbf{I}_{t-i}$, for $i = 1, 2, \cdots, N$. Specifically, each frame is partitioned into $K$ non-overlapping patches of size $p \times p$, yielding $\{\mathbf{I}_{t-i}^{(k)}\}_{k=1}^{K}$, where $k$ indexes the patches. For each patch, a binary mask $M_{t-i}^{(k)}$ is generated as follows: $M_{t-i}^{(k)} = 1$, if $\xi^{(k)} \geq r$; $M_{t-i}^{(k)} = 0$, if $\xi^{(k)} < r$. Here $\xi^{(k)} \sim \mathcal{U}(0,1)$ is a uniformly distributed random variable, and $r \in [0,1]$ is the patch drop rate controlling the probability of retaining each patch. The augmented frame $\tilde{\mathbf{I}}_{t-i}$ is then constructed by applying the masks to the corresponding patches: $\tilde{\mathbf{I}}_{t-i}^{(k)} = M_{t-i}^{(k)} \cdot \mathbf{I}_{t-i}^{(k)}$, for $k = 1, 2, \ldots, K$. This random omission of patches effectively disrupts detailed appearance information while preserving the coarse spatial structure necessary for modeling motion dynamics.

**Gaussian Noise Augmentation.** During the incremental generation process, previously generated video frames may introduce contamination in both appearance and dynamics, such as noise in facial regions and the background, or subtle distortions in lip movements and facial expressions. As this process continues, these contaminations can propagate to subsequent frames, leading to the gradual accumulation and amplification of artifacts. To mitigate this issue, we incorporate Gaussian noise into the motion frames, enhancing the denoiser's ability in the latent space to recover from contaminations in appearance and dynamics. Specifically, we introduce Gaussian noise to the augmented latent representations: $\hat{\mathbf{z}}_{t-i} = \tilde{\mathbf{z}}_{t-i} + \boldsymbol{\eta}_{t-i}, \boldsymbol{\eta}_{t-i} \sim \mathcal{N}(\mathbf{0}, \sigma^2 \mathbf{I})$, where $\sigma$ controls the noise level, and $\mathbf{I}$ denotes the identity matrix. The corrupted latent representations $\{\hat{\mathbf{z}}_{t-1}, \hat{\mathbf{z}}_{t-2}, \ldots, \hat{\mathbf{z}}_{t-N}\}$ are then used as motion condition inputs to the diffusion model.

| Method | FID↓ | FVD↓ | Sync-C↑ | Sync-D↓ | E-FID↓ |
|---|---|---|---|---|---|
| Audio2Head | 41.753 | 246.041 | **8.051** | **7.117** | 10.190 |
| SadTalker | 21.924 | 293.084 | 7.399 | 7.812 | 6.881 |
| EchoMimic | 47.331 | 532.733 | 5.930 | 9.143 | 11.051 |
| AniPortrait | 26.241 | 361.978 | 3.912 | 10.264 | 11.253 |
| Hallo | 16.748 | 366.066 | 7.268 | 7.714 | 7.081 |
| Ours | **16.616** | **239.517** | 7.379 | 7.697 | **6.702** |
| Real video | - | - | 8.377 | 6.809 | - |

Table 1: The quantitative comparisons with existed portrait image animation approaches on the HDTF dataset. Our evaluation focuses on generated videos with a duration of 4 minutes, maintaining consistent settings across subsequent quantitative experiments.

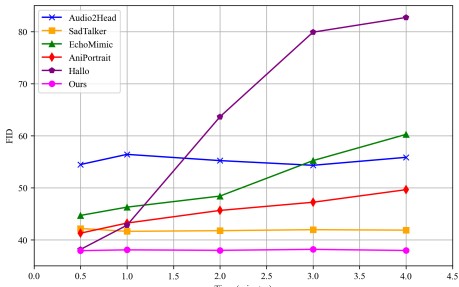

Figure 5: FID metrics of different methods as inference time increases.

These noise-augmented motion frames are incorporated into the diffusion process via cross-attention mechanisms within the denoising U-Net. At each denoising step $t$, the model predicts the noise component $\epsilon_\theta(\mathbf{z}_t, t, \mathbf{c})$, where $\mathbf{z}_t$ is the current noisy latent, and $\mathbf{c}$ represents the set of conditioning inputs: $\epsilon_\theta(\mathbf{z}_t, t, \mathbf{c}) = \epsilon_\theta\left(\mathbf{z}_t, t, \mathbf{z}_{\text{ref}}, \{\hat{\mathbf{z}}_{t-i}\}, \mathbf{c}_{\text{audio}}, \mathbf{c}_{\text{text}}\right)$. Here, $\mathbf{z}_{\text{ref}} = \mathcal{E}(\mathbf{I}_{\text{ref}})$ is the latent representation of the reference image, and $\mathbf{c}_{\text{audio}}, \mathbf{c}_{\text{text}}$ are the encoded audio features and textual embeddings, respectively. By leveraging the noise-augmented motion frames, the model effectively captures temporal dynamics while mitigating the influence of accumulated artifacts. This approach encourages that the subject's appearance remains stable, derived from the reference image, throughout the generated video sequence.

## 3.2 High-Resolution Enhancement

To enhance temporal coherence in high-resolution video generation, we adopt a codebook prediction approach Zhou et al. (2022c), incorporating an introduced temporal alignment mechanism.

As illustrated in Figure 4, we propose two methods for extracting input latent features. The first method directly uses latent features from the diffusion model. While straightforward, this approach requires end-to-end training of the entire super-resolution module. Alternatively, the second method processes the latent features through the diffusion model's decoder, followed by a low-quality encoder that maps them into a continuous latent space. This method leverages pretrained encoder, decoder, and codebook components, thereby simplifying the training process. Subsequently, code index prediction transformer module is then employed to predict the codebook index and select corresponding features from it, which are fed into a high-quality decoder to generate the super-resolution video. In order to keep the temporal consistency of super-resolution video, we integrate the temporal alignment within the code index prediction transformer module. Specifically, let $\mathbf{x}_{\text{self}} \in \mathbb{R}^{N \times (H \cdot W) \times C}$ denotes the hiddden state following the self-attention layer in the code prediction transformer module, where $N$ represents the frame number, $H$ and $W$ correspond to the height and width dimensions, and $C$ denotes the channel dimension. Subsequently, we reshape $\mathbf{x}_{self}$ into the shape of $(H \cdot W) \times N \times C$ and apply conventional scaled dot-product attention. By integrating temporal alignment, the network effectively captures intra-frame and inter-frame dependencies, enhancing both temporal consistency and visual fidelity in high-resolution video outputs.

## 3.3 Textual Prompt Control

To enable precise modulation of facial expressions and motions based on textual instructions, we incorporate an adaptive layer normalization mechanism into the denoising U-Net architecture. Given a text prompt, a text embedding $\mathbf{e}_{\text{text}}$ is extracted using the CLIP text encoder Radford et al. (2021). This embedding is processed through a zero-initialized multilayer perceptron (MLP) to produce scaling ($\gamma$) and shifting ($\beta$) parameters: $\gamma, \beta = \text{MLP}(\mathbf{e}_{\text{text}})$.

The adaptive layer normalization is applied between the cross-attention layer and the audio attention layer within the denoising U-Net. Specifically, the intermediate features $\mathbf{X}_{\text{cross}}$ from the cross-attention layer are adjusted as follows: $\mathbf{X}_{\text{norm}} = \text{LayerNorm}(\mathbf{X}_{\text{cross}})$, $\mathbf{X}_{\text{adapted}} = \gamma \odot \mathbf{X}_{\text{norm}} + \beta + \mathbf{X}_{\text{cross}}$, where $\odot$ denotes element-wise multiplication. This adaptation conditions the denoising process on the textual input, enabling fine-grained control over the synthesized expressions and motions in the generated video frames.

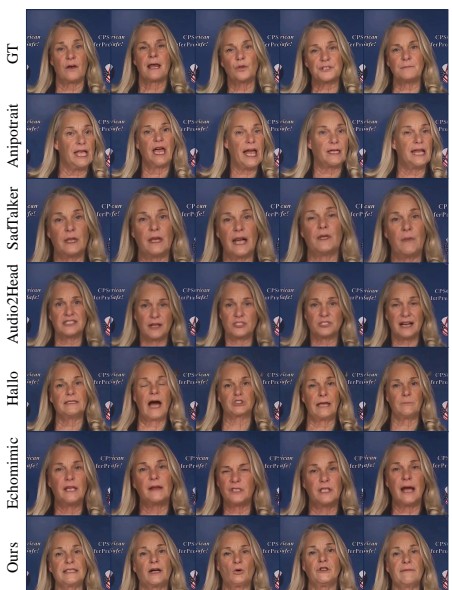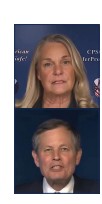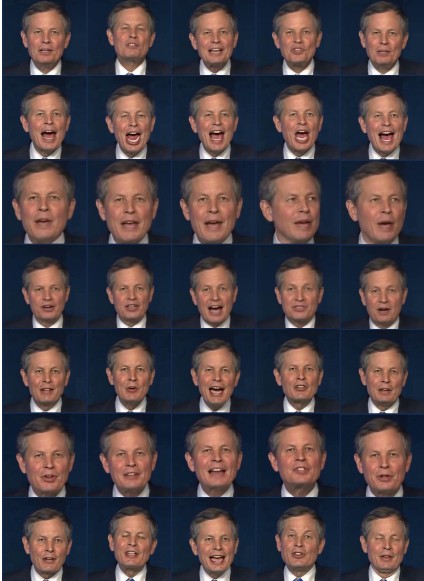

Reference
image

Figure 6: The qualitative comparison with exited approaches on HDTF data-set.

## 3.4 NETWORK

**Network Architecture.** Figure 3 illustrates the proposed approach's architecture. The ReferenceNet embeds the reference image $\mathbf{z}_{\text{ref}}$, capturing the visual appearance of both the portrait and the corresponding background. To model temporal dynamics while mitigating appearance contamination from preceding frames, the motion frames $\{\hat{\mathbf{z}}_{t-i}\}$ are subjected to patch dropping and Gaussian noise augmentation. Our extended framework utilizes a denoising U-Net architecture that processes noisy latent vectors $\mathbf{z}_t$ at each diffusion timestep $t$. The embedding of the input audio $\mathbf{c}_{\text{audio}}$ is derived from a 12-layer wav2vec network Schneider et al. (2019), while the textual prompt embedding $\mathbf{c}_{\text{text}}$ is obtained through CLIP Radford et al. (2021). By synthesizing these diverse conditioning inputs via cross-attention layers within the denoising U-Net Blattmann et al. (2023), the model generates frames that maintain visual coherence with the reference image while dynamically exhibiting nuanced and expressive lip motions and facial expressions. Finally, the high-resolution enhancement module employs vector quantization of latent codes in conjunction with temporal alignment techniques to produce final videos at 4K resolution.

## 4 EXPERIMENTS

### 4.1 EXPERIMENTAL SETUPS

**Implementation.** All experiments were conducted on a GPU server equipped with 8 NVIDIA A100 GPUs. The training process was executed in two stages: the first stage comprised 30,000 steps with a step size of 4, targeting a video resolution of $512 \times 512$ pixels. The second stage involved 28,000 steps with a batch size of 4, initializing the motion module with weights from Animatediff. Approximately 160 hours of video data were utilized across both stages, with a learning rate set at 1e-5. For the super-resolution component, training for temporal alignment was extended to 550,000 steps, leveraging initial weights from CodeFormer and a learning rate of 1e-4, using the VFHQ dataset as the super-resolution training data. Each instance in the second stage generated 16 video frames, integrating latents from the motion module with the first 4 ground truth frames, designated as motion frames. During inference, the output video resolution is increased to a maximum of 4096 × 4096 pixels.

Regarding textual control, we employed the vision-language model MiniCPM Hu et al. (2024) to generate textual prompts. These prompts were refined using Llama 3.1, which extracted expression and emotion descriptions from the original captions. The final textual prompt was formatted as {{human}{expression}}. During training, we randomly dropped the textual prompt condition with

| Method | FID↓ | FVD↓ | Sync-C↑ | Sync-D↓ | E-FID↓ |
|---|---|---|---|---|---|
| Audio2Head | 50.449 | 448.695 | 6.269 | 8.325 | 38.981 |
| SadTalker | 24.600 | 380.866 | 6.384 | 8.169 | 44.596 |
| EchoMimic | 50.994 | 854.826 | 5.082 | 9.675 | 35.806 |
| AniPortrait | 24.301 | 344.000 | 3.975 | 10.171 | 41.307 |
| Hallo | 28.186 | 571.991 | 6.610 | 8.181 | 36.793 |
| Ours | **24.072** | **360.192** | **6.760** | **8.156** | **33.316** |
| Real video | - | - | 7.088 | 7.726 | - |

Table 2: The quantitative comparisons with existed approaches on the proposed "Wild" data-set.

| Patch size | Drop rate | FID↓ | FVD↓ | Sync-C↑ | Sync-D↓ |
|---|---|---|---|---|---|
| 1 | 0.50 | 39.642 | 513.314 | 6.687 | 8.515 |
| 1 | 0.25 | 38.518 | **491.338** | **6.766** | 8.387 |
| 2 | 0.20 | 38.477 | 492.216 | 6.762 | 8.413 |
| 4 | 0.30 | **37.756** | 498.957 | 6.754 | **8.371** |
| 16 | 0.25 | 44.172 | 756.517 | 6.431 | 8.517 |

Table 3: The quantitative comparison of different settings of patch drop augmentation. The statistics are obtained on the CelebV dataset.

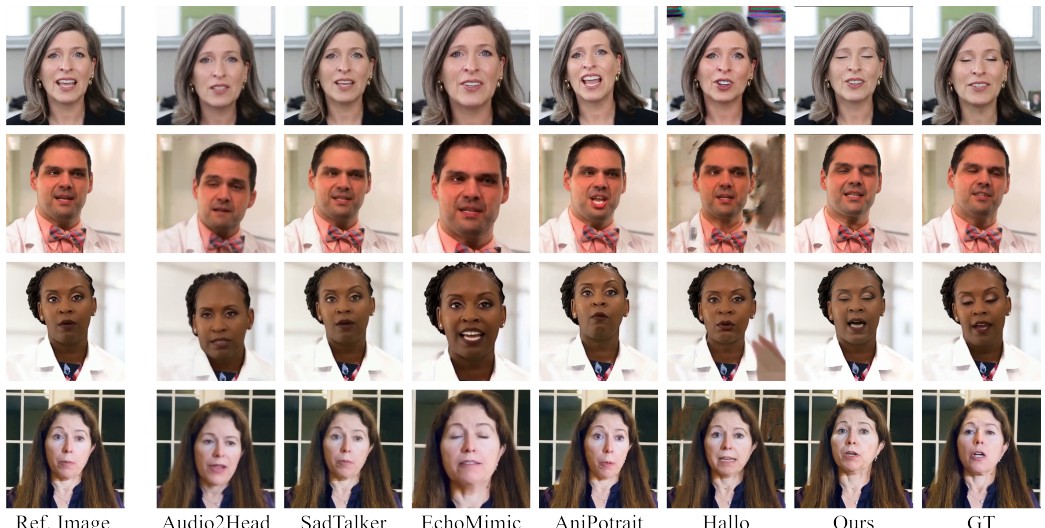

Ref. Image  Audio2Head  SadTalker  EchoMimic  AniPotrait  Hallo  Ours  GT

Figure 7: The qualitative comparison with existed approaches on the proposed "Wild" data-set.

a probability of 0.05. During inference, the classifier-free guidance (CFG) scale was uniformly set to 3.5 for the textual prompt, the reference image, and the audio. When comparing against other baseline methods, we used null textual prompt as the condition.

**Baseline Approaches.** We evaluate our framework against leading state-of-the-art techniques, including both non-diffusion and diffusion-based models. Non-diffusion models, such as Audio2Head and SadTalker, are compared with diffusion-based counterparts like EchoMimic, AniPortrait, and Hallo. Notably, EchoMimic and AniPortrait employ a parallel generation approach for long-duration outputs, while Hallo utilizes an incremental formulation. Unlike previous studies that focused on short-duration videos of only a few seconds, our evaluation is conducted on generated videos lasting 4 minutes, using looped audio from the benchmark dataset as the driving audio. To ensure a fair comparison, we have excluded the high-resolution enhancement module, maintaining the same output video resolution (512 × 512 pixels) as the existed approaches across all quantitative comparisons.

## 4.2 Comparison with State-of-the-Art

**Comparison on HDTF Dataset.** Table 1 and Figure 6 present quantitative and qualitative comparisons on the HDTF dataset. Our framework achieves the lowest FID of 16.616 and an E-FID of 6.702, demonstrating superior fidelity and perceptual quality. Additionally, our synchronization metrics, Sync-C (7.379) and Sync-D (7.697), further validate the effectiveness of our method. As illustrated in Figure 5, the extended inference duration significantly impacts FID metrics in existing diffusion-based approaches, leading to notable declines compared to their short-duration performance. In terms of lip and expression motion synchronization, parallel methods such as EchoMimic and AniPortrait exhibit marked deterioration. In contrast, our extended approach consistently demonstrates superior and stable performance across image and video quality, as well as motion synchronization, even as inference time increases.

**Comparison on the Proposed "Wild" Dataset.** Table 2 and Figure 7 offers additional quantitative and qualitative comparison results of the introduced "Wild" dataset. Our method achieves an FID of 24.072 and an E-FID of 33.316, both indicative of high image quality. We also register a Sync-C

| Gaussian noise | Patch drop | FID↓ | FVD↓ | Sync-C↑ | Sync-D↓ |
|---|---|---|---|---|---|
| | | 82.715 | 1088.158 | 6.683 | 8.420 |
| ✓ | | 78.283 | 984.876 | 6.701 | 8.415 |
| | ✓ | 38.518 | 491.338 | 6.766 | 8.387 |
| ✓ | ✓ | **37.944** | **477.412** | **6.928** | **8.307** |

Table 4: Ablation study of the patch drop and Gaussian noise augmentation on the CelebV data-set.

| Reference Image | Motion Frames | FID↓ | FVD↓ | Sync-C↑ | Sync-D↓ |
|---|---|---|---|---|---|
| | | 82.715 | 1088.158 | 6.683 | 8.420 |
| ✓ | | 98.374 | 1276.453 | 6.584 | 8.512 |
| ✓ | ✓ | 68.471 | 594.434 | 6.735 | 8.394 |
| | ✓ | **38.518** | **491.338** | **6.766** | **8.387** |

Table 5: Analysis of different combinations of patch augmentations applied to the reference image and motion frames.

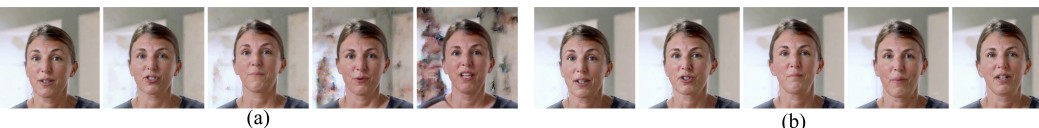

(a)      (b)

Figure 8: Effectiveness of Gaussian noise augmentation: (a) No augmentation is used for motion frames. (b) Gaussian noise augmentation is introduced to the motion frames.

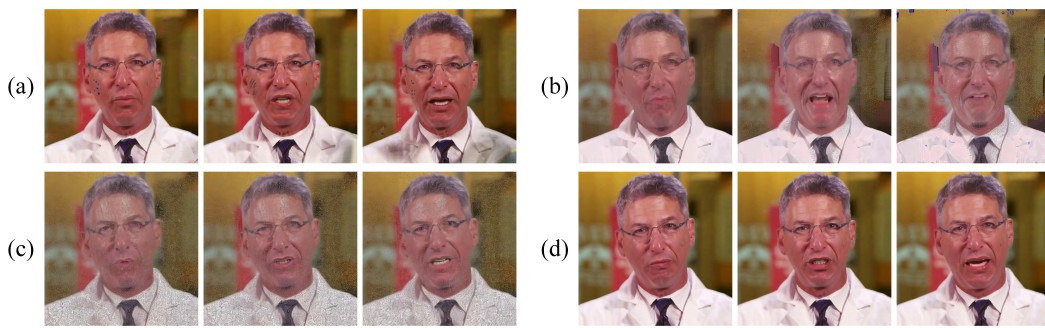

Figure 9: Different combinations of patch augmentation for the reference image and motion frames are as follows: (a) No patch-drop augmentation is used for either the reference or motion frames. (b) Patch-drop augmentation is introduced only to the reference image. (c) Patch-drop augmentation is added to both the reference image and motion frames. (d) Patch-drop augmentation is applied only to the motion frames.

score of 6.760 and a Sync-D of 8.156, alongside the highest FVD of 360.192, demonstrating superior coherent video structure.

## 4.3 ABLATION STUDIES AND DISCUSSION

**Different Patch Drop Sizes and Rate.** Table 3 presents a comparative analysis of varying patch sizes and drop rate applied to motion frames. Our results indicate that using patch sizes of 1–4 pixels with appropriate patch drop augmentation ratios of 20%–30% yields comparable performance on lip sync metrics (Sync-C and Sync-D) and image and video quality metrics (FID and FVD). Too large patch size or too large drop rate not only destroys the appearance details of motion frames, but also damages the facial motion of motion frames and reduces the quality of the generated video, resulting in a high FID and FVD scores.

**Effectiveness of Augmentation Strategies.** Table 4 evaluate different augmentation strategies. Gaussian noise alone results in a high FID of 82.715 and FVD of 1088.158, indicating suboptimal quality. As shown in Figure 8, without Gaussian noise augmentation, errors tend to accumulate in the background over time. In contrast, applying this augmentation strengthens the model's ability to suppress these errors, resulting in a clearer background. The patch drop strategy significantly improves these metrics, reducing FID to 38.518 and FVD to 491.338, as it effectively suppresses artifacts in portrait appearances, illustrated in Figure 9. Notably, the combined strategy further enhances performance, achieving the lowest FID of 37.944 and FVD of 477.412, alongside the highest Sync-C score of 6.928. Thus, the combined augmentation method proves to be the most effective in generating high-quality motion frames.

To assess the effectiveness of patch augmentation, we apply it to the reference and motion images, as presented in Table 5 and Figure 9. Our experiments demonstrate that without the proposed augmentation, the generated portraits (Figure 9(a)) exhibit numerous artifacts. Augmenting only the reference image contaminates its identity features (see Figure 9(b)) because noise accumulates across the motion frames. Similarly, applying augmentation to both the reference image and motion

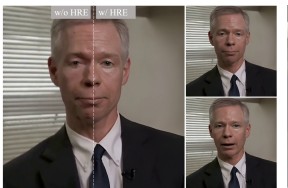 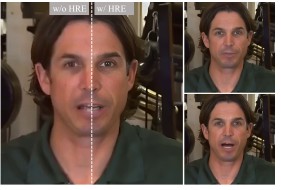

| Method | LPIPS↓ | FID↓ | FVD↓ | MUSIQ↑ |
|---|---|---|---|---|
| MagVITv2 | 0.2647 | 91.46 | 425.68 | 0.6841 |
| ESRGAN | 0.2532 | 88.78 | 418.06 | 0.7033 |
| Ours w/ HRE | **0.2310** | **66.92** | **360.71** | **0.7187** |

Figure 10: Comparison of the portrait image animation results with and without high-resolution enhancement.

Table 6: Quantitative evaluations of the super-resolution results from different methods. HRE represents the proposed high-resolution enhancement.

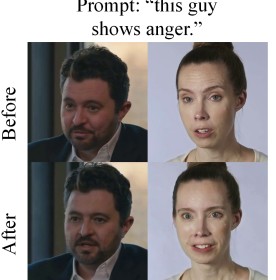 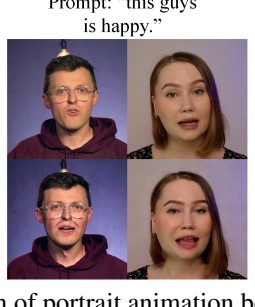 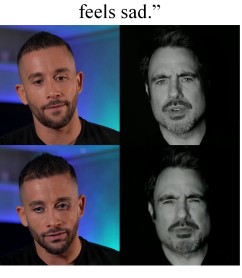 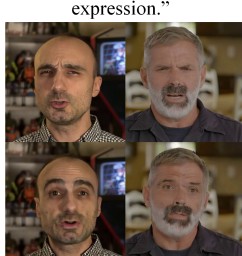

Figure 11: Comparison of portrait animation before and after applying the textual prompts.

frames introduces noise into the portraits (Figure 9(c)) due to the compromised reference image. To address this issue, we exclusively augment the motion frames, aiming for them to provide only facial motion while the reference image supplies the identity appearance. This approach as shown in Figure 9(d) preserves the identity appearance of the reference image while effectively capturing the facial motion.

**Effectiveness of High-Resolution Enhancement.** The effectiveness of high-resolution enhancement techniques is illustrated in Figure 10, which demonstrates improved animation quality via video super-resolution. We compare the proposed super-resolution method with previous models, specifically ESRGAN Wang et al. (2018) and MagVITv2 Yu et al. (2023). The corresponding statistics are provided in Table 6. The results indicate that although the introduced super-resolution approaches enhance visual details measured by MUSIQ, our proposed high-resolution enhancement (HRE) achieves superior video quality, as evidenced by better FVD scores. This improvement is due to the incorporation of the temporal alignment module, which enhances temporal coherence.

**Effectiveness of Textual Prompt.** The integration of textual prompts into our portrait image animation framework significantly enhances the control over generated animations, as illustrated in Figure 11. The comparative analysis demonstrates that textual prompts facilitate precise manipulation of facial expressions and emotional nuances, allowing for a more tailored animation output. By providing explicit instructions regarding desired emotional states, the model exhibits improved responsiveness in generating animations that align closely with the specified prompts.

## 5 CONCLUSION

This paper presents advancements in portrait image animation through the enhanced capabilities of the Hallo framework. By extending animation durations to tens of minutes while maintaining high-resolution 4K output, our approach addresses significant limitations of existing methods. Specifically, innovative data augmentation techniques, including patch-drop and Gaussian noise, ensure robust identity consistency and reduce appearance contamination. Furthermore, we implement vector quantization of latent codes and employ temporal alignment techniques to achieve temporally consistent 4K videos. Additionally, the integration of audio-driven signals with adjustable semantic textual prompts enables precise control over facial expressions and motion dynamics, resulting in lifelike and expressive animations. Comprehensive experiments conducted on publicly available datasets validate the effectiveness of our method, representing a significant contribution to the field of long-duration, high-resolution portrait image animation.

## 6 ACKNOWLEDGEMENTS

This project is sponsored by Natural Science Foundation of Shanghai under Grant No. 24ZR1407200 and Shanghai Oriental Talents Project under Grant No. QNKJ2024060.

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
