# OpenReview forum: "Hallo2: Long-Duration and High-Resolution Audio-Driven Portrait Image Animation"
_ICLR.cc/2025/Conference — ICLR 2025 Poster_

### Official Review · Reviewer_rhVg · 2024-10-31

**Soundness:** 3
**Presentation:** 3
**Contribution:** 2
**Rating:** 6
**Confidence:** 4

**Summary:**

This work presents an enhanced version of the prior work Hallo. This work improves upon the previous model in terms of long-term video generation, 4K high-resolution video generation, and semantic textual label condition. This work is the first one to study 4K audio-driven portrait image animation. Experiment results show that this method achieves state-of-the-art performance in long-term portrait video animation, which can generate rich and controllable 4K content of tens of minutes.

**Strengths:**

1. This is the first work which attempts to animate portrait videos under 4K resolution. The qualitative results shown in the supplementary material demonstrate the improvements against the baseline.
2. The one hour length portrait video animation is interesting. The Gaussian noise augmentation technique is effective in alleviating the error accumulation issue in the autoregressive long video animation process.
3.  The textual prompt condition helps maintain the emotion and expression of the talking person, which is simple yet effective.

**Weaknesses:**

1. This work is a combination of existing works. The autoregressive animation pipeline has already been explored in EMO. The noise simulation trick is commonly used in the autoregressive generation models to alleviate error accumulation. The 4K resolution generation is an extension of VQGAN in the audio-driven portrait animation. Therefore, this work is a combination of prior works from the engineering perspective, which is not novel or significant enough.
2. Compared with Hallo, the animation framework remains unchanged and is quite similar to prior works like EMO. The reference unet borrows ideas from human image animation but the authors didn’t cite them properly. Please consider cite [1,2] in the final version.
3. The long video results shown in the supplementary do not contain too many translation and rotation of the talking head, which cannot demonstrate the temporal consistency for long-term animation. It is recommended to conduct such an experiment with diverse translation and rotation of the talking head to prove the effectiveness of this proposed method.
The ablation of HRE is not comprehensive enough. Comparisons with the 512x512 model cannot show the necessity of developing such a VQGAN-based superresolution in the pipeline. It is recommended to compare the proposed method with previous superresolution models, such as ESRGAN[3].

[1] Animate Anyone: Consistent and Controllable Image-to-Video Synthesis for Character Animation
[2] MagicAnimate: Temporally Consistent Human Image Animation using Diffusion Model
[3] Real-ESRGAN: Training Real-World Blind Super-Resolution with Pure Synthetic Data

**Questions:**

No questions, please see the weaknesses for rebuttal.

---

> ### Author Response · Authors · 2024-11-27
> **Official Comment by Authors (1)**
>
> **[Q1]: This work is a combination of existing works. The autoregressive animation pipeline has already been explored in EMO. The noise simulation trick is commonly used in the autoregressive generation models to alleviate error accumulation. The 4K resolution generation is an extension of VQGAN in audio-driven portrait animation. Therefore, this work is a combination of prior works from the engineering perspective, which is not novel or significant enough.**
>
> **[A1]**: Our work is built based on existing works, and advances the portrait image animation area in duration and resolution. In particular, our work can generate 1 hour video with identity maintained and background consistent, which is not reached by previous approaches.
>
> In terms of methods, (1) we agree that noise simulation trick is comonlly used, but it still is not able to generate long (e.g., 1 hour) video. Our dropping pixel scheme is simple, not explored for portrait image animation, and works surprisingly well. (2) we extend codeformer to video superresolution, by adding the temporal alignment.
>
> In summary, we believe that our work benefits the portrait image animation area, especially generating high-quality and consistent long videos.
>
> **[Q2]: Compared with Hallo, the animation framework remains unchanged and is quite similar to prior works like EMO. The reference unet borrows ideas from human image animation but the authors didn’t cite them properly. Please consider cite [1,2] in the final version.**
>
> **[A2]**: The updated manuscript cites two significant works (Line 157-158) that greatly contribute to our research: Animate Anyone and MagicAnimate.
>
> Animate Anyone introduces a novel framework for character animation based on diffusion models. It features ReferenceNet, which utilizes spatial attention to preserve intricate appearance details, a pose guider for controlling character movements, and a temporal modeling approach to ensure smooth inter-frame transitions.
> Meanwhile, MagicAnimate presents a diffusion-based framework that enhances human image animation by improving temporal consistency and identity preservation. This is achieved through an innovative video diffusion model and appearance encoder.
>
> Both frameworks demonstrate state-of-the-art quantitative results on benchmark datasets and effectively animate arbitrary characters with impressive results.

---

> ### Author Response · Authors · 2024-11-27
> **Official Comment by Authors (2)**
>
> **[Q3]: The long video results shown in the supplementary do not contain too many translation and rotation of the talking head, which cannot demonstrate the temporal consistency for long-term animation. It is recommended to conduct such an experiment with diverse translation and rotation of the talking head to prove the effectiveness of this proposed method.**
>
> **[A3]**: Following the recommendation, we conducted experiments of talking head with diverse translation and rotation. Hallo and ours indicate the statistics testing on HTDF data-sets. We obtain the diverse translations and rotations of the talking head (results of the last row) by applying a higher coefficient on pose, as introduced in Hallo.
>
> We observe that with diverse translation and rotation of the talking head, our method maintains comparable performance in image and video quality metrics (FID and FVD) and lip synchronization accuracy (Sync-C and Sync-D), compared with the original reported results of ours.
>
> |  |$\qquad$**FID ↓**|$\quad\qquad$**FVD ↓**|$\quad\qquad$**Sync-C ↑**|$\qquad$**Sync-D ↓** |
> |:-|:-|:-|:-|:-|
> |Hallo|$\space\quad$16.748|$\qquad$366.066|$\space\quad\qquad$7.268|$\space\qquad$7.714|
> |Ours|$\space\quad$**16.616**|$\qquad$**239.517**|$\space\quad\qquad$**7.379**|$\space\qquad$**7.697**|
> |Ours\*|$\space\quad$16.697|$\qquad$249.301|$\space\quad\qquad$7.302|$\space\qquad$7.706|
>
> (Ours* represents the talking head with diverse translation and  rotation.)
>
> **[Q4]: The ablation of HRE is not comprehensive enough. Comparisons with the 512x512 model cannot show the necessity of developing such a VQGAN-based superresolution in the pipeline. It is recommended to compare the proposed method with previous superresolution models, such as ESRGAN.**
>
> **[A4]**: We follow the kind suggestion and compare the proposed method with previous superresolution models on samples of CelebV, specifically ESRGAN. The corresponding statistics are updated in Table 6 of the revision. The results show that both ESRGAN and our method enhance visual details, as measured by MUSIQ. However, our proposed high-resolution enhancement (HRE) achieves superior video quality, as reflected by FVD scores.
>
> |  | $\qquad$**LPIPS ↓** | $\space\qquad$**FID ↓**  | $\space\space\qquad$  **FVD ↓**   |$\qquad$**MUSIQ ↑** |
> |:-|:-|:-|:-|:-|
> | ESRGAN                |$\qquad$0.2532  |  $\qquad$88.78   |$\qquad$  418.06 | $\space\qquad$0.7033  |
> | Ours w/HRE         |$\qquad$**0.2310**  |$\qquad$**66.92**   |  $\qquad$ **360.71** |$\qquad$ **0.7187**  |
>
> (HRE represents the proposed high-resolution enhancement)

---

> > ### Author Response · Authors · 2024-11-29
> >
> > Dear Reviewer,
> >
> > We hope our response has addressed your questions. As the discussion phase is coming to a close, we are looking forward to your feedback and would like to know if you have any remaining concerns we can address. We are grateful if you find our revisions satisfactory and consider raising your score for our paper.
> >
> > Thank you once again for the time and effort you have dedicated to reviewing our paper.
> >
> > Best regards!
> >
> > Hallo2 Authors

---

> > > ### Comment · Reviewer_rhVg · 2024-12-01
> > >
> > > Thanks for the detailed responses. I have carefully read the comments and responses of other reviewers. Most of my concerns have been addressed by the rebuttal, so I raised my rating to 6. The reason for not considering the higher score is that I think the novelty of this work is limited as it is built on top of existing works, which is also mentioned by the authors in the response.

---

> > > > ### Author Response · Authors · 2024-12-01
> > > >
> > > > We are delighted to receive your response and suggestions. Thank you for raising your evaluation and for your support.

---

### Official Review · Reviewer_a9qA · 2024-11-01

**Soundness:** 3
**Presentation:** 4
**Contribution:** 1
**Rating:** 6
**Confidence:** 5

**Summary:**

This paper proposes the first method to generate an hour-long, audio-driven portrait image animation at 4K resolutions coupled with textual prompts. To accomplish this goal, the paper applies patch-drop augmentation on Gaussian noise, a vector quantization on latent codes with a high-quality decoder, and uses enhanced textual labels as the conditional inputs. These improvements constitute the proposed contributions.

**Strengths:**

The paper offers several contributions that are deemed to be the strengths of this paper. This work is an extension of a previous work on Hallo, thus termed Hallo2. It still attempts to propose a few enhancements that can further address the limitations of the previous work, mainly by extending it to long-hour portrait image animation with high resolutions.

The patch-drop data augmentation looks interesting. It intentionally adds corruption to the appearance of the conditional frames while preserving the motion information. Meanwhile, the appearance contamination is resolved using the Gaussian noise as the other data augmentation.

The 4K resolution was achieved by leveraging the VQ-GAN and code sequence prediction (existing but applied to a new task) to ensure smooth transitions among the temporal frames.

Therefore, the main strengths of this paper would lie in the patch-drop augmentation, gaussian noise augmentation and the code sequence prediction.

**Weaknesses:**

The main idea of patch-drop data augmentation is to separate the motion from the appearance, where the former is controlled by temporal dynamics in the preceding frames. The patch-augmentation was applied to individual latent images z_{t-i}, where it was further partitioned into the K non-overlapping patches. Each patch was randomly masked. The question here is if we generate random masking and apply that to the time step t-1, would the same masking positions be enforced with the t-2 to t-N? Or the t-2 to t-N time steps still use a random masking operation. Could you clarify that and also show the motivation or the merits of each design? The other question is that it makes sense the omission of patches can still preserve the coarse structure. The motion dynamics on the other hand are not affected by the masking operations. Could you provide more details on why this is the case?

Another intriguing part is related to the choice of patch size for dividing the latent image. Based on the results given in Table 3, the best patch size is 1*1, which means that a single pixel is dropped. This seems to contradict the intuition of using a patch. I don't mean 1 pixel is not a patch, but it does not conform to the normal choice of patch size. Could you offer more insights on why 1 pixel is considered to be more effective in this task? This would be very important.

Regarding the preceding generated frames, which approach was used? Somehow, I am not able to find that detail in the paper, or maybe I miss it. Was that achieved by generating the second frame based on the first one, and then continually generating the third frame based on the first two frames?

The design of Gaussian noise augmentation is simple and easy to understand, which was implemented by adding the Gaussian noise to the augmented latent representations. This was used as one of the conditional inputs. Could you provide more insights on why such an operation works? Ensuring temporal consistency is difficult. The paper seems to offer a much simpler and non-complex solution. Therefore, it would be more necessary to provide stronger validations on such a design.

The set of conditional inputs used in this work includes four parts, which are the audio, textual, noise-augmented motion frames, and reference image encoding. Based on the flowchart given in Figure 3, the cross-attention conditions occur in different layers of the encoder and decoder of the U-Net. Could you justify how to select the corresponding layer for the designated condition? Are they all implemented via cross-attention?

The high-resolution enhancement is still unclear to me. The paper does not seem to apply the super-resolution module but relies on the output from the diffusion model’s decoder and then a low-quality decoder. The technical part description in Sec 3.2 looks more like normal QKV operations. How the 4K resolution was achieved is still unclear, nor it was depicted in Figure 3.

The other issue might be related to the over-claim of the proposed approach by saying the "hour-long". The evaluation was done on minutes duration (e.g., 4 minutes in evaluation). More evidences are required for this "hour-long" claim.

The textual prompt implementation looks just like a regular conditional injection. This paper claims it to be one of the strengths of this work.

The paper needs to add some more comparisons such as the EMO work. This has been mentioned in the Introduction but appears to be not compared in the experimental section.

**Questions:**

Please carefully address the above questions. I think there are many details that need to be clarified, particularly in the individual components constituting the proposed method.

---

> ### Author Response · Authors · 2024-11-27
> **Official Comment by Authors (1)**
>
> **[Q1]: The question here is if we generate random masking and apply that to the time step t-1, would the same masking positions be enforced with the t-2 to t-N? Or the t-2 to t-N time steps still use a random masking operation. Could you clarify that and also show the motivation or the merits of each design?**
>
> **[A1]**: In our implementation, the t-2 to t-N time steps use the same masking operation in our work. We follow kind suggestions and add a comparison shown in the following table to clarify the metrics of each design. The comparison indicates that using random masking at each time step and using the same masking positions across all time steps yield comparable results in terms of visual quality and lip synchronization metrics. The motivation of each design both aims to have the generated video's facial identity appearance rely more on the reference image and less on the motion frames.
>
> |  |$\quad\quad$**FID ↓**|$\quad\quad$**FVD ↓**|$\quad$**Sync-C ↑**|$\quad$**Sync-D ↓** |
> |:-|:-|:-|:-|:-|
> |random masking|$\quad$**16.608**|$\quad$242.761|$\quad$7.368|$\quad$7.713|
> |same masking|$\quad$16.616|$\quad$**239.517**|$\quad$**7.379**|$\quad$**7.697**|
>
> **[Q2]: The other question is that it makes sense the omission of patches can still preserve the coarse structure. The motion dynamics on the other hand are not affected by the masking operations. Could you provide more details on why this is the case?**
>
> **[A2]**: The paper indicates that motion dynamics—including lip movements, facial expressions, and poses—primarily rely on coarse structural information. For instance, partial contours of the eyes, nose, lips, and overall facial outlines are sufficient to represent these dynamics. In contrast, facial identity appearance requires all the fine-grained features. As the reviewer mentioned, the omission of patches can still preserve the coarse structure. Therefore, by appropriately omitting patches (with a drop rate of 20%–30%, as determined in our following findings), the essential motion dynamics information can still be effectively retained.
>
> **[Q3]: Another intriguing part is related to the choice of patch size for dividing the latent image. Based on the results given in Table 3, the best patch size is 1*1, which means that a single pixel is dropped. This seems to contradict the intuition of using a patch. I don't mean 1 pixel is not a patch, but it does not conform to the normal choice of patch size. Could you offer more insights on why 1 pixel is considered to be more effective in this task? This would be very important.**
>
> **[A3]**: Our experimental results show that using bigger patch sizes (1–4 pixels) with appropriate patch drop ratios (20%-30%) yields satisfactory and comparable performance on lip sync metrics (Sync-C and Sync-D) and image and video quality metrics (FID and FVD). We add additional statistics to the following table, and update Table 3 of the updated revision.
> Specifically, for motion frames, dropping smaller patches allows the generation process to more effectively leverage facial identity information from the reference image, while preserving essential contour details related to facial motion—such as the eyes, nose, and mouth.
>
> | **Patch size** | $\qquad$**Drop rate** | $\qquad$**FID ↓** | $\qquad$**FVD ↓**   | $\qquad$**Sync-C ↑** | $\qquad$**Sync-D ↓** |
> |:----:|:---:|:------:|:-----:|:------:|:------:|
> | 1              | $\qquad$0.50          | $\quad$39.642   | $\space$$\quad$513.314    |$\quad$6.687       |$\quad$8.515       |
> | 1              |$\qquad$0.25          |$\quad$38.518   |$\quad$ **491.338** |$\quad$**6.766**   |$\quad$8.387       |
> | 2              |$\qquad$0.20          |$\quad$38.477   |$\space$$\quad$492.216    |$\quad$6.762       |$\quad$8.413       |
> | 4              |$\qquad$0.30          |$\quad$**37.756**|$\space$$\quad$498.957    |$\quad$6.754       |$\quad$**8.371**   |
> | 16             |$\qquad$0.25          |$\quad$44.172   |$\space$$\quad$756.517    |$\quad$6.431       |$\quad$8.517       |
>
> **[Q4]: Regarding the preceding generated frames, which approach was used? Was that achieved by generating the second frame based on the first one, and then continually generating the third frame based on the first two frames?**
>
> **[A4]**: On the first time of inference, we duplicate the reference image to create two identical motion frames. These frames are used to generate a video sequence of 16 frames. After that, we take the last two frames of this sequence as new motion frames to generate the next set of frames. We do this operation iteratively until generating all frames.

---

> ### Author Response · Authors · 2024-11-27
> **Official Comment by Authors (2)**
>
> **[Q5]: The design of Gaussian noise augmentation is simple and easy to understand, which was implemented by adding the Gaussian noise to the augmented latent representations. This was used as one of the conditional inputs. Could you provide more insights on why such an operation works? Ensuring temporal consistency is difficult. The paper seems to offer a much simpler and non-complex solution. Therefore, it would be more necessary to provide stronger validations on such a design.**
>
> **[A5]**: The effectiveness of Gaussian noise augmentation lies in its ability to simulate the error accumulation process in motion frames. By introducing Gaussian noise to augmented latent representations, the model learns to recover from motion-induced errors and generate consistent and realistic portrait videos.
> As demonstrated in Figure 8 of the revised paper, without Gaussian noise augmentation, errors tend to accumulate in the background over time. With Gaussian noise augmentation, the model effectively suppresses these errors, resulting in clearer and more consistent video background.
>
> **[Q6]: The set of conditional inputs used in this work includes four parts, which are the audio, textual, noise-augmented motion frames, and reference image encoding. Based on the flowchart given in Figure 3, the cross-attention conditions occur in different layers of the encoder and decoder of the U-Net. Could you justify how to select the corresponding layer for the designated condition? Are they all implemented via cross-attention?**
>
> **[A6]**: We have updated Figure 3 in the revised submission to clarify the role of input conditions in video generation.
> Specifically, the input audio is encoded using Wav2Vec and projected through an MLP to generate key and value representations, while the latent features are encoded as queries. Cross-attention is then applied to incorporate the audio conditioning. For the reference image features, $z_{\text{ref}}$ is concatenated with the latent features $z_{\text{t}}$ and integrated via self-attention. Additionally, the textual prompt is encoded using CLIP and incorporated through adaptive layer normalization. Noise-augmented motion frames are encoded by reference net, and the output features are concatenated with the latent features $z_{\text{t}}$ and integrated via self-attention in the motion module.
>
> **[Q7]: The high-resolution enhancement is still unclear to me.**
>
> **[A7]**: In Section 3.2 of the revised submission (Line 274-282), we follow the kind suggestion and clarify the high-resolution enhancement process. Following option #2 in Figure 4, we first decode the low-resolution latent features and use a low-quality encoder to map them into a continuous latent space. We then apply spatial and temporal attention mechanisms and perform code index prediction to select corresponding features from the codebook. Finally, these selected features are fed into a high-quality decoder to generate the super-resolution video.
>
> **[Q8]: The other issue might be related to the over-claim of the proposed approach by saying the "hour-long". The evaluation was done on minute duration (e.g., 4 minutes in evaluation). More evidence is required for this "hour-long" claim.**
>
> **[A8]**: We follow the suggestion and provide the "hour-long" video demo via the following anonymous link:
> https://1drv.ms/v/s!AueTKT2aVehrgQd654p4-jhuA3Gt?e=bsnPeK .
>
> **[Q9]: The textual prompt implementation looks just like a regular conditional injection. This paper claims it to be one of the strengths of this work.**
>
> **[A9]**: The method of textual prompt condition injection is regular, but we can generate animation videos with diverse expressions with the help of textual prompt (please see Figure 11), which is different from previous works.
>
> **[Q10]: The paper needs to add some more comparisons such as the EMO work. This has been mentioned in the Introduction but appears to be not compared in the experimental section.**
>
> **[A10]**: Based on the videos and corresponding audio (arround 1 minutes) provided on the official EMO website, we conducted a comparative analysis. We found that, given the same reference image and driving audio, both methods exhibit comparable lip synchronization capabilities, while our approach achieves superior image detail quality, measured by MUSIQ.
> Our approach builds upon EMO and other notable Unet-based diffusion models for portrait image animation, and further enhances long-duration and high-resolution (please see the table below) video generation.
>
> | |$\quad$**Sync-C ↑**|$\quad$**Sync-D ↓**|$\quad$**MUSIQ ↑**|
> |:-|:-:|:-:|:-|
> |EMO|$\quad$4.30|$\quad$8.99|$\quad$0.6316|
> |Ours|$\quad$**4.36**|$\quad$**8.92**|$\quad$**0.7117**|

---

> > ### Author Response · Authors · 2024-12-01
> >
> > Dear Reviewer,
> >
> > We sincerely appreciate your valuable feedback and the time you've dedicated to reviewing our paper. As the extended discussion phase nears its end, we kindly request any final feedback you may have on our revisions. Please let us know if there are any remaining issues we can address. We hope our revisions have satisfactorily addressed your concerns, and we respectfully ask that you consider raising your evaluation of our paper.
> >
> > Thank you once again for your time and effort.
> >
> > Best regards,
> >
> > The Hallo2 Authors

---

> > > ### Comment · Reviewer_a9qA · 2024-12-02
> > >
> > > Thanks for addressing the raised concerns. There are several responses which are still not very convinced. Regarding the response to Q3, where the choice of patch size was chosen. The conclusion looks more like based on the empirical evaluation. It still does not address on why patch size 1*1 is more effective in this case. Regarding Q8 with this hour-long video demo, this is just a single subject with relatively controlled pose, where the audio and lip movements are relatively synchronized. It will be more interesting to see demonstrated evidence to support this hour-long video generation with quantitive metrics as this has been highly claimed in this paper. Regarding Q9, it still looks like the textual prompt is just a regular implementation, with a slightly different application. It is unclear whether this could be claimed as one of the strengthes.

---

> > > > ### Author Response · Authors · 2024-12-02
> > > >
> > > > **1. Regarding the response to Q3, where the choice of patch size was chosen. The conclusion looks more like based on the empirical evaluation. It still does not address on why patch size 1*1 is more effective in this case.**
> > > >
> > > > **[A1]**: Thanksａlot! The reason for that patch size 1\*1 is more effective is that: size 1*1 is able to preserve the motion information. For a large size, a patch containing the motion information (eg., lip movements, facial expressions, and poses) might be dropped, and accordingly, maintaining the temporal consistency does not get enough information from motion frames, increasing the training difficult. In summary, using patch size 1\*1 is able to keep the motion information and at the same time to "destroy" the appearance in motion frames to make the network learn appearance from reference image.
> > > >
> > > > **2. Regarding Q8 with this hour-long video demo, this is just a single subject with relatively controlled pose, where the audio and lip movements are relatively synchronized. It will be more interesting to see demonstrated evidence to support this hour-long video generation with quantitive metrics as this has been highly claimed in this paper.**
> > > >
> > > > **[A2]**: Thank you for your suggestion. We will add experiments to support the hour-long video generation with quantitative metrics. Since these experiments require some time, we hope to provide the results later today. Thank you again.
> > > >
> > > >
> > > > **3. Regarding Q9, it still looks like the textual prompt is just a regular implementation, with a slightly different application. It is unclear whether this could be claimed as one of the strengthes.**
> > > >
> > > > **[A3]**: Thanks for your great advice. We will follow your suggestion, and make this part as an application other than claiming it as a strength.

---

> > > > > ### Author Response · Authors · 2024-12-02
> > > > > **Results of the additional experiments**
> > > > >
> > > > > We follow the reviewer's suggestion and conducted additional experiments to support hour-long video generation, accompanied by quantitative metrics. The relevant quantitative results were evaluated on 8 subjects selected from the CelebV dataset.
> > > > >
> > > > > We present 3 tables containing quantitative metrics:
> > > > >
> > > > >
> > > > > Table 1: Quantitative comparisons with existing portrait image animation approaches for a duration of 10 minutes.
> > > > > |               |$\space\quad$**FID ↓**|$\space\space\quad$**FVD ↓**|$\quad$**Sync-C ↑**|$\quad$**Sync-D ↓**|$\quad$**E-FID ↓**|
> > > > > |:---------------|:-------|:--------|:--------|:--------|:-------|
> > > > > | Audio2Head    | $\quad$68.12 | $\quad$515.65 | $\space\space\quad$6.11   | $\space\space\quad$**8.27**   | $\quad$61.23 |
> > > > > | SadTalker     | $\quad$49.56 | $\quad$520.67 | $\space\space\quad$6.07   | $\space\space\quad$8.34   | $\quad$28.46 |
> > > > > | EchoMimic     | $\quad$69.86 | $\quad$843.93 | $\space\space\quad$5.40   | $\space\space\quad$8.71   | $\quad$22.72 |
> > > > > | AniPortrait   | $\quad$57.01 | $\quad$568.76 | $\space\space\quad$2.75   | $\space\quad$13.33  | $\quad$26.31 |
> > > > > | Hallo         | $\quad$93.77 | $\space\space\space$1141.42| $\space\space\quad$6.02   | $\space\space\quad$8.61   | $\quad$17.86 |
> > > > > | Ours          | $\quad$**41.21** | $\quad$**487.17** | $\space\space\quad$**6.39**   | $\space\space\quad$8.33   | $\quad$**15.57** |
> > > > >
> > > > >
> > > > >
> > > > > Table 2: Quantitative comparisons with existing portrait image animation approaches for a duration of 60 minutes.
> > > > > |               |$\space\quad$**FID ↓**|$\space\space\quad$**FVD ↓**|$\quad$**Sync-C ↑**|$\quad$**Sync-D ↓**|$\quad$**E-FID ↓**|
> > > > > |:---------------|:-------|:--------|:--------|:--------|:-------|
> > > > > | Audio2Head    | $\quad$78.27 | $\quad$531.93 | $\space\space\quad$6.10   | $\space\space\quad$**8.29**   | $\quad$61.76 |
> > > > > | SadTalker     | $\quad$57.21 | $\quad$580.92 | $\space\space\quad$6.02   | $\space\space\quad$8.30   | $\quad$29.62 |
> > > > > | EchoMimic     | $\quad$75.69 | $\quad$864.71 | $\space\space\quad$5.37   | $\space\space\quad$8.94   | $\quad$30.07 |
> > > > > | AniPortrait   | $\quad$66.84 | $\quad$585.43 | $\space\space\quad$2.73   | $\space\quad$13.27  | $\quad$32.43 |
> > > > > | Hallo         | $\space\space\space$102.70| $\space\space\space$1155.21| $\space\space\quad$5.95   | $\space\space\quad$8.76   | $\quad$21.84 |
> > > > > | Ours          | $\quad$**41.54** | $\quad$**490.48** | $\space\space\quad$**6.35**   | $\space\space\quad$8.39   | $\quad$**15.91** |
> > > > >
> > > > > Table 3: Quantitative metrics of our approach for video generation results of varying time durations.
> > > > > |               |$\space\quad$**FID ↓**|$\space\space\quad$**FVD ↓**|$\quad$**Sync-C ↑**|$\quad$**Sync-D ↓**|$\quad$**E-FID ↓**|
> > > > > |----------------|-------|--------|--------|--------|-------|
> > > > > | 1 minute       | $\quad$40.83 | $\quad$486.11 | $\space\space\quad$6.37   | $\space\space\quad$8.32   | $\quad$14.89 |
> > > > > | 10 minute      | $\quad$41.21 | $\quad$487.17 | $\space\space\quad$6.39   | $\space\space\quad$8.33   | $\quad$15.57 |
> > > > > | 30 minute      | $\quad$42.53 | $\quad$487.39 | $\space\space\quad$6.34   | $\space\space\quad$8.32   | $\quad$15.34 |
> > > > > | 60 minute      | $\quad$41.54 | $\quad$490.48 | $\space\space\quad$6.35   | $\space\space\quad$8.39   | $\quad$15.91 |
> > > > >
> > > > > From these experiments, we can see that as the inference time increases and even to 60 minutes, our proposed method demonstrates stable and superior performance in both visual quality and lip synchronization.

---

### Official Review · Reviewer_MzNt · 2024-11-02

**Soundness:** 3
**Presentation:** 3
**Contribution:** 2
**Rating:** 6
**Confidence:** 4

**Summary:**

This paper proposes Hallo2 for audio-driven face image animation, which mainly includes three training strategies. Firstly, this paper proposes a patch-drop data augmentation technique to prevent contamination of appearance information from preceding frames. Secondly, this paper extends the Vector Quantized Generative Adversarial Network discrete codebook space method for code sequence prediction tasks into the temporal dimension. Finally, this paper also introduces additional textual prompts for semantic control.

**Strengths:**

1. This paper discusses how to generate high-quality animated videos driven by audio, which is a very interesting topic.
2. The structure of this paper is well-organized and easy to follow.
3. The experimental results show the effectiveness of the proposed method.

**Weaknesses:**

There are some questions and suggestions,
1. This paper proposes three strategies to improve the performance of audio-driven face image animation; however, the motivation for each strategy is not well explained. For example, in Section 3.2, a codebook prediction approach is used to enhance temporal coherence. It would be helpful if the authors provided more insights into why temporal coherence is insufficient without this codebook prediction approach.
2. This paper presents three distinct strategies; however, the connections between each contribution are weak, making it difficult for reviewers to grasp the central contribution. This lack of coherence also gives the impression that each contribution is only incremental.
3. From lines 238 to 241, the author uses a masking strategy to mask frames in the video. The author claims that this approach improves consistency. However, I’m a bit puzzled. In video generation, the masking strategy is generally beneficial for generating long videos autoregressively, so why would masking improve consistency? Could you please give more insights?
4. Some descriptions in the HIGH-RESOLUTION ENHANCEMENT section are a bit confusing. In the first half of this section, it seems that a Transformer is used to model the video in a continuous latent space. However, Figure 4 actually shows a codebook. Since it appears the authors are using a codebook, I suggest reorganizing the first half of this section to reflect this.
5. I observed flickering in some of the videos in the experimental results, especially with the man wearing a checkered shirt. Could this be due to limitations in the pre-trained VAE? Given that there is temporal alignment in the HIGH-RESOLUTION ENHANCEMENT section, why hasn’t this temporal alignment helped reduce the flickering effect?

**Questions:**

Please see above. If the author solves my problems, I will consider raising the score. Thanks.

---

> ### Author Response · Authors · 2024-11-27
> **Official Comment by Authors (1)**
>
> **[Q1]: This paper proposes three strategies to improve the performance of audio-driven face image animation; however, the motivation for each strategy is not well explained. For example, in Section 3.2, a codebook prediction approach is used to enhance temporal coherence. It would be helpful if the authors provided more insights into why temporal coherence is insufficient without this codebook prediction approach.**
>
> **[A1]**: The motivation for the codebook prediction approach in Section 3.2 is to enhance the super-resolution of facial visual details. We acknowledge that making frame-level independent predictions without temporal constraints can lead to inconsistencies across the temporal dimension in the generated 4K images. To mitigate this issue, we introduce codebook prediction with temporal alignment enhancement, aiming to ensure visual consistency over time following super-resolution.
> As demonstrated in the table below, the application of codebook prediction (without temporal alignment) based super-resolution (second row) results in obvious improvement in image detail quality, measured by MUSIQ metric. However, we observed that temporal coherence between video frames remained insufficient, indicated by suboptimal FVD metrics. By incorporating temporal alignment into the codebook prediction process (third row), we maintain enhanced super-resolution results while also substantially improving temporal coherence across frames.
>
> | Code Prediction | $\qquad$Temporal Alignment | $\quad\qquad$**MUSIQ ↑** | $\quad\qquad$**FVD ↓**  |
> |:----:|:-------:|:----:|:-----:|
> |    |       | $\space$$\qquad$0.6797  |$\qquad$365.423|
> |    √    |   |$\qquad$ 0.7089  |$\qquad$378.548|
> | √     | √     | $\qquad$ **0.7187**  | $\qquad$**360.712**|
>
> **[Q2]: From lines 238 to 241, the author uses a masking strategy to mask frames in the video. The author claims that this approach improves consistency. However, I’m a bit puzzled. In video generation, the masking strategy is generally beneficial for generating long videos autoregressively, so why would masking improve consistency? Could you please give more insights?**
>
> **[A2]**: Before applying the masking strategy, reference images contained face identity appearance information, while motion frames included both facial motion and identity appearance details. As the generated videos become longer, noise and distortions in the face identity appearance from the motion frames can accumulate. This accumulation leads to the face identity appearance in the generated video gradually deviating from that of the reference image, resulting in inconsistency (see first row of the following table). Please see the first row of the following table (Table 5 and Figure 9 of the updated revision).
> By introducing the masking strategy, we encourage the generated video to rely primarily on the reference images for face identity appearance, minimizing dependence on the motion frames for appearance details. Since the motion frames are intended to provide motion information only, masking their appearance features ensures that the face identity appearance in the generated video comes almost entirely from the reference image. This approach enhances the consistency of the face identity appearance throughout the generated video. Please see the last row of the following table.
>
> | Reference Image | $\qquad$ Motion Frames | $\qquad$ **FID ↓**   | $\quad\qquad$**FVD ↓**     | $\quad\qquad$**Sync-C ↑** | $\quad\qquad$**Sync-D ↓** |
> |:----------------:|:-------------:|:------:|:--------:|:-------:|:-------:|
> |                 |               | $\qquad$82.715 | $\qquad$1088.158 | $\space$$\qquad$6.683   | $\space$$\qquad$8.420   |
> | ✓               |               | $\qquad$98.374 | $\qquad$1276.453 | $\space$$\qquad$6.584   |$\qquad$ 8.512   |
> | ✓               | ✓             | $\qquad$68.471 | $\qquad$594.434  |$\qquad$ 6.735   | $\space$$\qquad$8.394   |
> |                 | ✓             | $\qquad$**38.518** | $\qquad$**491.338** |$\space$$\qquad$**6.766** | $\space$$\qquad$**8.387** |

---

> ### Author Response · Authors · 2024-11-27
> **Official Comment by Authors (2)**
>
> **[Q3]: Some descriptions in the HIGH-RESOLUTION ENHANCEMENT section are a bit confusing. In the first half of this section, it seems that a Transformer is used to model the video in a continuous latent space. However, Figure 4 actually shows a codebook. Since it appears the authors are using a codebook, I suggest reorganizing the first half of this section to reflect this.**
>
> **[A3]**: Yes! We have followed the kind suggestion and reorganized Section 3.2 to clarify our approach, as reflected in the revised manuscript ( Line 274-282).
> In Section 3.2, we first decode the low-resolution latent features. Then, we use a low-quality encoder to map these features into a continuous latent space. After applying spatial and temporal attention mechanisms, we perform code index prediction. The predicted indices are then passed into the codebook for super-resolution operations, as illustrated in Figure 4.
>
> **[Q4]: I observed flickering in some of the videos in the experimental results, especially with the man wearing a checkered shirt. Could this be due to limitations in the pre-trained VAE? Given that there is temporal alignment in the HIGH-RESOLUTION ENHANCEMENT section, why hasn’t this temporal alignment helped reduce the flickering effect?**
>
> **[A4]**:This is due to the limitation of the pretrained VEA. The temporal alignment can partially reduce the effect, but can't  handle it satisfactorily. Because the latent feature encoded by VAE loses some information due to perceptual compression, making it difficult to accurately reconstruct the original content during decoding. This limitation results in flickering in the reconstructed video frames, which is particularly noticeable when handling checkered details such as the patterns on the man’s shirt. As shown in the table below, we demonstrate the statistics of the checkered shirt video with and without temporal alignment.
>
> |  |$\qquad$**LPIPS ↓**|$\quad\qquad$**FID ↓**|$\quad\qquad$**FVD ↓**|$\qquad$**MUSIQ ↑** |
> |:-|:-|:-|:-|:-|
> |without temporal alignment|$\qquad$0.1397|$\qquad$**39.098**|$\qquad$368.298|$\qquad$0.7451|
> |with temporal alignment|$\qquad$**0.1395**|$\qquad$39.112|$\qquad$**341.673**|$\qquad$**0.7587**|

---

> > ### Author Response · Authors · 2024-11-29
> >
> > Dear Reviewer,
> >
> > We hope our response has addressed your questions. As the discussion phase is coming to a close, we are looking forward to your feedback and would like to know if you have any remaining concerns we can address. We are grateful if you find our revisions satisfactory and consider raising your score for our paper.
> >
> > Thank you once again for the time and effort you have dedicated to reviewing our paper.
> >
> > Best regards!
> >
> > Hallo2 Authors

---

> > > ### Comment · Reviewer_MzNt · 2024-11-29
> > >
> > > Thank you for your reply. Although each proposed strategy is relatively scattered and has no progressive relationship, considering that the author has solved some of my doubts, the video generated from audio and face is also relatively good. I graded the score to 6. For the generation of audio2video, I have not read much about the existing literature. And **we need to refer to other reviewers for its novelty judgment**.

---

> > > > ### Author Response · Authors · 2024-12-01
> > > >
> > > > Thank you for your response and for enhancing your evaluation of our work. We are delighted to receive your suggestions and implementing them in both revision and future work.

---

### Official Review · Reviewer_VuDm · 2024-11-03

**Soundness:** 3
**Presentation:** 3
**Contribution:** 3
**Rating:** 6
**Confidence:** 4

**Summary:**

This paper presents a pipeline for generating long-duration, high-resolution, audio-driven facial animations. To enable extended video generation, the authors introduce patch-drop and Gaussian noise augmentation techniques during training. For high-resolution output, the method incorporates a VQ-based super-resolution stage and temporal alignment attention to ensure temporal consistency. Additionally, a norm-based textual prompt control mechanism is used to enhance the expressiveness of facial animations.

Experiments across multiple datasets demonstrate the effectiveness of the proposed approach.

**Strengths:**

- An effective augmentation strategy  which do not need to change model structure and support long-duration

- The textual condition injection propose a novel method to inject text prompt to a reference-net based animation structure.

**Weaknesses:**

- For high-resolution enhancement, the paper lacks a comparison with video VQ models (e.g., MagVITv2) and does not include quantitative evaluations of the super-resolution results from the SR encoder-decoder.

- For textual prompt control, the paper does not provide a detailed description of how prompts are used in comparisons with baseline methods or how prompts are incorporated during training data preparation. Additionally, details regarding the CFG process for managing multiple conditions (e.g., text, audio, reference image) are missing.

**Questions:**

- For patch augmentation, could the authors analyze the “effective augmentation ratio”? Since the purpose of patch augmentation is to encourage the model to disregard appearance information and focus on motion, an analysis of which patches contain motion information and which contain appearance information, along with a calculation of the effective augmentation ratio for the video clip, would provide valuable insight into the effectiveness of this design.

- How is the textual condition used during inference? Since it serves as an additional conditioning factor, the authors should clarify how this information is labeled, details of training behavior (e.g., random dropping for CFG), and how CFG is applied during inference. It would also be useful to include an analysis of the impact of conflicts between non-textual and text-audio conditions.

- On the HDTF dataset (i.e., Table 1), the proposed method outperforms most methods except Audio2Head. Could the authors provide an explanation for this? The paper should include further explanation of the evaluation metrics used to help readers understand which capabilities are being measured and improved.

- In the third image on the bottom right side of Figure 6, there appear to be “water droplet-like” artifacts. Could the authors explain the cause of these artifacts?

---

> ### Author Response · Authors · 2024-11-27
> **Official Comment by Authors (1)**
>
> **[Q1]: For high-resolution enhancement, the paper lacks a comparison with video VQ models (e.g., MagVITv2) and does not include quantitative evaluations of the super-resolution results from the SR encoder-decoder.**
>
> **[A1]:** We appreciate your insightful suggestion. In Table 6 of the updated revision, we have included a comparison with video VQ models including MagVITv2 and include quantitative evaluations of the super-resolution results from the SR encoder-decoder.
> It's important to note that various VQ models and super-resolution methods are viable; we have selected one representative approach for this study. We are willing to follow the MagVITv2 formulation for future research.
> |$\quad$**Method**  | $\quad\qquad$ **LPIPS ↓** | $\quad\qquad$ **FID ↓**  |  $\quad\qquad$ **FVD ↓**   |$\quad\qquad$**MUSIQ ↑** |
> |:-:|:-:|:-:|:-:|:-:|
> | MagVITv2| $\qquad$ 0.2647|$\qquad$ 91.46|$\space\space\qquad$ 425.68|$\qquad$ 0.6841|
> | ESRGAN                |$\qquad$ 0.2532  |  $\qquad$ 88.78   | $\space\space\qquad$ 418.06 | $\qquad$ 0.7033  |
> | Ours w/HRE         |$\qquad$ **0.2310**  | $\qquad$ **66.92**   | $\space\space\qquad$  **360.71** | $\qquad$ **0.7187**  |
>
> (HRE represents the proposed high-resolution enhancement)
>
> **[Q2]: Since textual prompt serves as an additional conditioning factor, the authors should clarify how this information is labeled, details of training behavior (e.g., random dropping for CFG). How prompts are incorporated during training data preparation.**
>
> **[A2]:** To obtain the textual prompt in training data preparation, we use a vision-language model, named MiniCPM, to generate initial textual prompts. After that, we obtain the final textual prompt using Llama 1.5 to extract the key words containing expression and emotion from the original full caption by MimiCPM and format it. The textural prompt has the following format: {{human}{expression}}.
> As for details of training behavior, we train our model using almost 160 hours of videos without the textual control module first. After that, we add the textual control module to our model and only train this textual control module using about 40 hours of videos with textual prompt. During training, we use a probability of 0.05 to random drop the textual prompt condition.
>
> **[Q3]: For textual prompt control, the paper does not provide a detailed description of how prompts are used in comparisons with baseline methods. How is the textual condition used during inference?**
>
> **[A3]**: The detailed description of how prompts are used is as follows and updated in Line 405-411 of revision. During inference, there are two different ways to apply the textual condition. First, we use null textual prompt as the condition, when comparing with other baseline methods.  Second, we use the regular textual prompt describing human emotion and set the CFG scale of textual prompt to 3.5 when discussing the effectiveness of textual prompt.
>
> **[Q4]: How CFG is applied during inference？It would also be useful to include an analysis of the impact of conflicts between non-textual and text-audio conditions.**
>
> **[A4]**: During inference, we set the CFG scale for the textual prompt, reference image, and audio all to 3.5. We also conducted an experiment to analyze the impact of conflicts between conditions. By comparing the first and third rows of the following table, we observe that a higher reference image CFG scale results in better image quality, as measured by FID. Similarly, comparing the second and third rows shows that a higher audio CFG scale improves lip synchronicity, as measured by Sync-C and Sync-D. Furthermore, comparing the third and the fourth rows reveals that a higher textual prompt CFG scale leads to more vivid expressions, as measured by E-FID. To balance all conditions and generate high-quality video, we set the CFG scale for the textual prompt, reference image, and audio all to 3.5.
>
> | $\quad$**Image** | $\qquad$ **Audio** | $\qquad$ **Text** | $\qquad$ **FID ↓** | $\qquad$ **FVD ↓**  | $\qquad$ **Sync-C ↑** | $\qquad$ **Sync-D ↓** | $\qquad$ **E-FID ↓** |
> |:----:|:----:|:----:|:----:|:----|:----:|:----:|:----:|
> |  6         |$\qquad$3.5       | $\qquad$ 3.5      |$\qquad$**24.47**    |$\qquad$260.07    | $\qquad$7.51        |$\qquad$8.06        |$\qquad$7.998      |
> | 3.5       |$\qquad$6         |$\qquad$3.5      |$\qquad$27.03    |$\qquad$247.87    |$\qquad$**8.04**        |$\qquad$**7.69** |$\qquad$9.257      |
> | 3.5       |$\qquad$3.5       |$\qquad$3.5      |$\qquad$27.04    |$\qquad$240.40    |$\qquad$7.87        |$\qquad$7.81        |$\qquad$**7.596**      |
> | 3.5       |$\qquad$3.5       |$\qquad$1.5      |$\qquad$25.21    |$\qquad$**235.41**   |$\qquad$7.88        |$\qquad$7.78        |$\qquad$7.624      |

---

> ### Author Response · Authors · 2024-11-27
> **Official Comment by Authors (2)**
>
> **[Q5]: For patch augmentation, could the authors analyze the “effective augmentation ratio”? Since the purpose of patch augmentation is to encourage the model to disregard apearance information and focus on motion, an analysis of which patches contain motion information and which contain appearance information, along with a calculation of the effective augmentation ratio for the video clip, would provide valuable insight into the effectiveness of this design.**
>
> **[A5]**: To analyze the "effective augmentation ratio" we conducted experiments with different patch sizes and drop rate settings, as presented in Table 3 of the updated paper. Our results indicate that using patch sizes of 1–4 pixels with appropriate patch drop augmentation ratios of 20%–30% yields comparable performance on lip sync metrics (Sync-C and Sync-D) and image and video quality metrics (FID and FVD).
> The paper indicates that motion dynamics include lip movements, facial expressions, and poses. Therefore, patches that include contours of the eyes, nose, lips, and overall facial outlines contain motion information.  Meanwhile, appearance information—which comprises both facial identity related details and background texture information—is inherently present in all patches.
>
> | **Patch size** | $\qquad$**Drop rate** | $\qquad$**FID ↓** | $\qquad$**FVD ↓**   | $\qquad$**Sync-C ↑** | $\qquad$**Sync-D ↓** |
> |:----:|:---:|:------:|:-----:|:------:|:------:|
> | 1              | $\qquad$0.50          | $\quad$39.642   | $\space$$\quad$513.314    |$\quad$6.687       |$\quad$8.515       |
> | 1              |$\qquad$0.25          |$\quad$38.518   |$\quad$ **491.338** |$\quad$**6.766**   |$\quad$8.387       |
> | 2              |$\qquad$0.20          |$\quad$38.477   |$\space$$\quad$492.216    |$\quad$6.762       |$\quad$8.413       |
> | 4              |$\qquad$0.30          |$\quad$**37.756**|$\space$$\quad$498.957    |$\quad$6.754       |$\quad$**8.371**   |
> | 16             |$\qquad$0.25          |$\quad$44.172   |$\space$$\quad$756.517    |$\quad$6.431       |$\quad$8.517       |
>
> **[Q6]: On the HDTF dataset (i.e., Table 1), the proposed method outperforms most methods except Audio2Head. Could the authors provide an explanation for this?**
>
> **[A6]**: While Audio2Head outperforms our method in the Sync-C and Sync-D metrics—which assess the synchronization between audio and lip movements—this is due to its use of explicit facial keypoints, enabling more precise, parameterized modeling of lip movements. However, reliance on these keypoints may lead to generated outputs that lack realism in facial expressions. In contrast, our approach demonstrates a significant advantage over Audio2Head in the FID, FVD, and EFID metrics, reflecting superior visual quality and more realistic facial expressions.
>
>
> **[Q7]: The paper should include further explanation of the evaluation metrics used to help readers understand which capabilities are being measured and improved.**
>
> **[A7]**: The explanation of the evaluation metrics are provided as below. Specifically, FID measures the similarity between the distribution of generated images and real images. It evaluates the quality of generated images by computing the Fréchet distance between their feature distributions in a high-dimensional space.  FVD is the video counterpart of FID, designed to evaluate the quality of generated videos. Sync-C and Sync-D metrics evaluate the synchronization between audio and lip movement in generated videos by caiculating the similarity between audio feature and vision feature extracting by syncnet. E-FID evaluates the expressiveness of the facial expressions in the generated videos by compute the FID score of generated video expression parameters and ground-truth video expression parameters.
>
> **[Q8]: In the third image on the bottom right side of Figure 6, there appear to be “water droplet-like” artifacts. Could the authors explain the cause of these artifacts?**
>
> **[A8]**: We have replaced the incorrect figure with the proper one in the updated revision.

---

> > ### Author Response · Authors · 2024-12-01
> >
> > Dear Reviewer,
> >
> > We hope that our responses have thoroughly addressed your questions. As the discussion phase comes to a close, we eagerly await your feedback and would appreciate knowing if you have any remaining concerns we can address. We would be grateful if you find our revisions satisfactory and consider raising your evaluation of our paper.
> >
> > Thank you once again for the time and effort you have dedicated to reviewing our manuscript.
> >
> > Best regards,
> >
> > The Hallo2 Authors

---

> > > ### Comment · Reviewer_VuDm · 2024-12-02
> > >
> > > Thank you for the detailed response from the authors. The reply addresses my questions regarding the inference process, the impact of patch augmentation, and demonstrates the effectiveness of high-resolution enhancement. However, as the paper focuses on high-resolution and long-duration video generation, some water drop-let like artifacts, alias, and background flickering are still noticeable in the generated video. Therefore, I will maintain the score at 6.

---

### Meta-Review · Area_Chair_ZYzM · 2024-12-21

**Metareview:**

The paper addresses the problem of audio-driven portrait image animation and proposes a series of improvements which enable long duration video, high resolution (4k), and incorporate adjustable semantic textual labels for portrait expressions. The long duration video is enabled through a new patch-drop technique. The high resolution is done through vector quantization of latent codes as well as a high quality decoder. The paper also introduces a new "Wild" dataset.

The core strength of the paper is the strong qualitative results and the very high resolution and very long duration generations. The main weakness appears to be the somewhat incremental nature of the paper.

I agree with all reviewers and advocate for acceptance believing that the strengths outweigh the weaknesses.

**Additional Comments On Reviewer Discussion:**

All reviewers advocate for acceptance, four 6 ratings. Reviewers generally praised the strong results, though the potentially limited technical novelty (as the paper is a "combination of existing works") was noted (rhVg). Concerns about the precise benefits from each additional component were resolved during rebuttal (VuDm, MzNt)

---

### Decision · Program_Chairs · 2025-01-22

Accept (Poster)